# DROSIA: Decoupled Representation on Sequential Information Aggregation for Time Series Forecasting

## Abstract

Time series forecasting is crucial in various fields, including finance, energy consumption, weather, transportation, and network traffic. It necessitates effective and efficient sequence modeling to encapsulate intricate temporal relationships. However, conventional methods often aggregate sequential information into representations of each time point by considering other points in the sequence, thereby ignoring the intra-individual information and suffering from inefficiency. To address these challenges, we introduce a novel approach, **DROSIA**: **D**ecoupled **R**epresentation **O**n **S**equential **I**nformation **A**ggregation, which only integrates temporal relationships once as an additional representation for each point, achieving sequential information aggregation in a decoupled fashion. Thus balancing between individual and sequential information, along with a reduction in computational complexity. We select several widely used time series forecasting datasets, and previously top-performing models and baselines, for a comprehensive comparison. The experimental results validate the effectiveness and efficiency of DROSIA, which achieves state-of-the-art performance with only linear complexity. When provided with a fair length of input data, the channel-independent DROSIA even outperforms the current best channel-dependent model, highlighting its proficiency in sequence modeling and capturing long-distance dependencies. Our code will be made open-source in the subsequent version of this paper.

## 1 Introduction

A time series is a sequence of data points recorded in chronological order, which reflects the attribute characteristics of an object at various stages of its dynamic development. Time series data spans across numerous fields, including finance, energy consumption, weather, transportation, and network traffic. This type of data typically presents high-dimensional features and long sequences, characterized by intricate nonlinear relationships between time points. These complexities make it challenging to predict future developments accurately based on historical data. Consequently, time series forecasting stands as one of the most significant and challenging domains within data analysis, demanding effective and efficient sequence modeling to capture complex temporal relationships.

In recent years, numerous studies on time series forecasting have shown that deep learning methods significantly outperform traditional approaches, elevating deep learning forecasters to the forefront of research. For example, MLP-based models (Oreshkin et al., 2020; Tolstikhin et al., 2021; Zeng et al., 2023; Li et al., 2023; Zhang et al., 2022; Han et al., 2024) have garnered significant interest for their simplicity, efficiency, and predictive accuracy. CNN-based (Bai et al., 2018; Wang et al., 2022; Gao et al., 2020; Sen et al., 2019; Liu et al., 2022; Wu et al., 2023) and RNN-based (Lai et al., 2018; Voelker et al., 2019; Salinas et al., 2020) models have enhanced forecasting effectiveness by integrating local or global spatio-temporal information from time series data. Subsequently, methods based on attention mechanism have emerged as the dominant approach in sequence modeling, empowering numerous deep learning forecasters (Qin et al., 2017) to further refine their temporal relationship capturing capabilities. Particularly, Transformer-based models (Li et al., 2019; Chen et al., 2021; Zhou et al., 2021; Liu et al., 2021; Zhou et al., 2022; Zhang & Yan, 2023; Nie et al., 2023; Liu et al., 2024; Dai et al., 2024), have showcased unparalleled prowess in sequence modeling.

Existing sequence modeling methods typically aggregate sequential information into representations of each time point by considering other points in the sequence, which overlooks the unique information within individual points and may lacks efficiency. For instance, the self-attention mechanism, attends to all time points to update the current one, leading to a quadratic computational complexity that can become a bottleneck in the training and the inference processes (Dao et al., 2022). Additionally, the distinct information within each point can be compromised during sequence modeling. However, "the 'structure' (sedimented individual meanings) is powerful" (Fine, 1993). Inspired from the concept of Transverse Interaction: Individuals recognize the physical environment as a symbolic other and use this understanding to structure their interaction with a "generalized other" (Weigert, 1991). we propose a sociological perspective on the relationship between time series and individual points, which emphasizes that individual information is of great significance and necessitates a full interaction with the collective to enhance sequence modeling. Current methods, however, may overly sacrifice individual information for the sake of sequential information.

To illustrate our concept and address the limitations of current sequence modeling methods, we have developed a novel approach called DROSIA, which integrates rich temporal relationships as additional representations for each time point, thereby enhancing the expressive power of the data and better balancing the trade-off between sequential information and individual point information. We have conducted comprehensive experiments on several prominent and frequently used multivariate long-term time series forecasting datasets. DROSIA has demonstrated exceptional sequence modeling capabilities, and the results suggest that our proposed model attains state-of-the-art performance in downstream tasks while notably decreasing computational complexity. The contributions of this paper can be summarized as follows:

- We propose a novel sequence modeling method – DROSIA, which aggregates sequential information in a decoupled fashion, effectively balancing it with information of individuals.

- DROSIA exhibits exceptional proficiency in time series forecasting, achieving state-of-the-art performance with linear complexity, especially in experiments involving long sequences and large datasets, highlighting its efficacy in capturing long-distance dependencies.

- When compared to several previous state-of-the-art channel-dependent models, DROSIA demonstrates superior performance across all datasets with a fair input length compared to the channel amount. Note that DROSIA does not leverage any inter-channel information.

## 2   RELATED WORK

**Sequential Information Aggregation Methods.** Sequence information aggregation, or sequence modeling, is a pivotal technology across various fields, including natural language processing, speech recognition, and time series analysis. RNNs (Elman, 1990) process sequential information through recursive computations. LSTM (Hochreiter & Schmidhuber, 1997) and GRU (Cho et al., 2014) are two most commonly employed variants, which effectively manage the forgetting and retention of information via gating mechanisms, thereby mitigating the challenges traditional RNNs encounter when learning long-distance dependencies. RCNN (Girshick et al., 2014; Gu et al., 2021) leverages the strengths of both RNNs and CNNs (LeCun et al., 1998), extracting local features through convolutional operations before aggregating information via recursive computations.

Subsequently, the attention mechanism has become the dominant technology for sequence modeling. Traditional models have been bolstered by the integration of attention mechanisms (Qin et al., 2017), and the Transformer (Vaswani et al., 2017), which is built on self-attention, has seen remarkable success across a wide range of tasks. However, the attention mechanism has drawbacks in terms of computational efficiency. Its high computational cost can be a significant barrier for many researchers and engineers, thereby hindering its widespread adoption and dissemination.

**Time Series Forecasting Models.** In recent years, deep networks have advanced significantly in time series forecasting. RNN-based models (Lai et al., 2018; Voelker et al., 2019; Salinas et al., 2020) are effective in capturing temporal relationships but suffer from computational inefficiency and limited capability in modeling long-distance dependencies. CNN-based models (Bai et al., 2018; Wang et al., 2022; Gao et al., 2020; Sen et al., 2019; Liu et al., 2022; Wu et al., 2023), which perform convolution to hierarchically extract temporal features, have achieved competitive forecasting performance. MLP-based models (Oreshkin et al., 2020; Tolstikhin et al., 2021; Zeng

et al., 2023; Li et al., 2023; Zhang et al., 2022; Yi et al., 2023; Wang et al., 2024a; Han et al., 2024) have garnered considerable interest due to their efficient data processing and ability to capture temporal relationships.

Inspired by the capabilities of Transformer-based models (Li et al., 2019; Chen et al., 2021; Zhou et al., 2021; Liu et al., 2021; Zhou et al., 2022; Zhang & Yan, 2023; Nie et al., 2023; Liu et al., 2024; Dai et al., 2024) in capturing long-distance dependencies and complex temporal relationships, they have been extensively applied across various time series tasks. Prior research has largely centered on point-wise modeling. However, due to the computational complexity of Transformer, numerous studies have sought to enhance efficiency. The PatchTST (Nie et al., 2023) has demonstrated the advantages of representing time series through patching, effectively reducing sequence length while boosting forecasting performance. Nevertheless, Transformer-based methods still struggle with efficiency in multivariate long-term prediction scenarios. iTransformer (Liu et al., 2024) approaches the problem by representing each channel as a whole along the time axis and applying the Transformer encoder to these representations, which significantly reduces complexity but at the cost of losing temporal information, leading to suboptimal performance in cases with fewer channels and longer sequences. TimeXer (Wang et al., 2024b) leverages the benefits of both PatchTST and iTransformer, achieving promising results, yet the computational time remains a significant drawback.

Moreover, current research related to large language model (LLM) has attracted significant interest. Numerous researchers leverage the pre-trained LLMs to time series analysis (Zhou et al., 2023; Sun et al., 2024), including the forecasting (Chang et al., 2023; Gruver et al., 2023; Pan et al., 2024; Jin et al., 2024). Benefiting from the vast amount of pre-trained data and the well-structured embedding space, the LLM-based forecasters have demonstrated promising performance in time series forecasting tasks. LLM4TS (Chang et al., 2023) and "OneFitAll" (Zhou et al., 2023) finetune the LLMs to align the original word embedding with time series embeddings, While TEST (Sun et al., 2024), $S^2$IP-LLM (Pan et al., 2024), and TIME-LLM (Jin et al., 2024) tokenize the time series data first, and align them to the semantic space of LLMs, then enhance the models' effectiveness through various prompt techniques. However, some researchers have also questioned the effectiveness of LLM-based methods in time series forecasting (Tan et al., 2024), after conducting thorough experiments for LLM and non-LLM forecasters, they claimed that "despite the recent popularity of LLMs in time series forecasting, they do not appear to meaningfully improve performance".

## 3 METHODOLOGY

Time series can be defined as $X = \{x_1, x_2, \ldots, x_t\}, x \in \mathbb{R}^d$, where $t$ represents the current time point, starting from 1, and $d$ denotes the dimensionality of the features at each time point. The objective of time series forecasting is to predict the sequence $Y = \{x_{t+1}, x_{t+2}, \ldots, x_{t+h}\}$, with $h$ being the prediction horizon. We propose a novel method called **D**ecoupled **R**epresentation **O**n **I**nformation **A**ggregation, abbreviated as DROSIA, which comprises three components: patch embedding, DROSIA encoding, and linear decoding. The overall architecture is depicted in Figure 1.

### 3.1 PATCH EMBEDDING

To enhance prediction accuracy and computational efficiency, we define a sliding window of length $k$ as $T_i = \{x_{i+1}, x_{i+2}, \ldots, x_{i+k}\}$ with a stride of $s$, to segment the time series into patches. We utilize a fully connected linear layer for patch embedding, which takes each patch as input and produces a single vector as the patch's representation, referred to as $S_i$ for the $i$-th patch.

$$S_i = Linear(T_i), i = 1, 2, \ldots, n \tag{1}$$

In Equation (1), $n$ represents the total number of input patches. The linear layer treats the multivariate time series as multiple univariate series (in a channel-independent manner) (Han et al., 2023), multiplying the $k$ values within the sliding window by a matrix with dimensions $k \times d$, where $d$ is the dimensionality of patch embedding. This approach aligns with the methodology of PatchTST (Nie et al., 2023). which has been shown the advantages in long-term time series forecasting tasks across various related studies. After extracting patch-wise representations from the time series, PatchTST utilizes the Transformer (Vaswani et al., 2017) to encode these embeddings. In contrast, we employ DROSIA as the encoder. The following subsection will delve into the implementation details of DROSIA and highlight its distinctions from the self-attention mechanism and other methods.

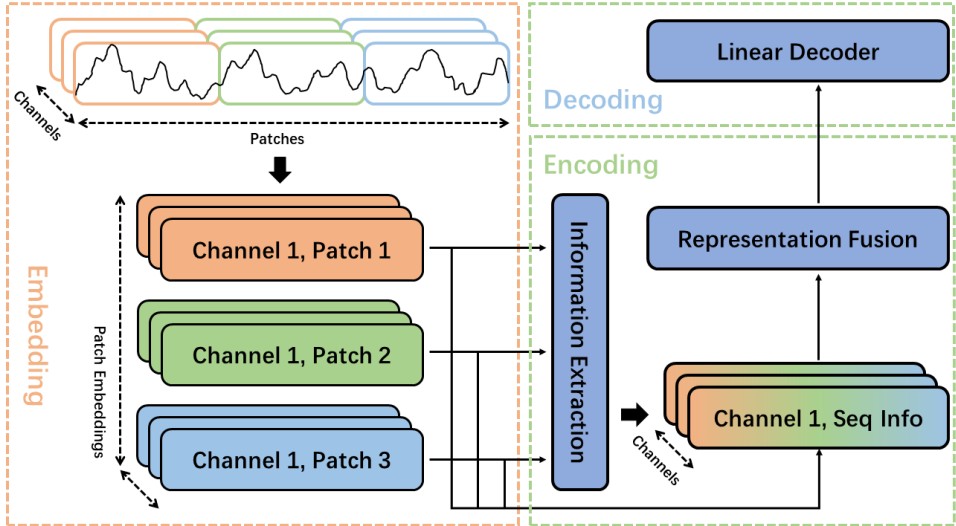

Figure 1: Overall architecture of DROSIA model, includes patch embedding, DROSIA encoding, and linear decoding. Note that DROSIA encoder could be repeatedly used. Information extraction extracts sequential information from all patch embeddings in the same channel, and fuses it with these embeddings in a decoupled manner. We will describe the details in the following of this paper.

## 3.2 DROSIA ENCODING

The DROSIA encoding module extracts sequential information from the patch embeddings, serving as additional representations of these patches, and then fuses the two back to original dimensionality of embeddings. In multi-layer networks, this process can be repeated, indicating that the fused representation can either be passed through another encoding layer or directly input into the decoder.

$$S^{j+1} = DROSIA(S^j), j = 1, 2, \ldots, l \tag{2}$$

Equation (2) outlines the overall process of the DROSIA encoder, which will be described in detail from Equation (3) to Equation (7). In this context, $DROSIA$ refers to a single encoder layer, $l$ denotes the number of layers. $S^1$ indicates the input to the first encoder layer, meanwhile the output of the embedding layer. $S^j$ is the input to the $j$-th layer. We consider sequential information as additional representation of the input, to achieve representation decoupling. The encoder primarily comprises three stages: sequence aggregation, information extraction, and representation fusion.

**Sequence Aggregation.** The output representations from the patch embedding or the previous layer of DROSIA encoder are first concatenated, which we refer to as sequence aggregation.

$$C^j = S_i^j \circ S_{i+1}^j \circ \cdots \circ S_{i+k}^j \tag{3}$$

In Equation (3), the $\circ$ represents the concatenate operation. The high-dimensional representation resulting from this concatenation is rich in temporal information, which must be fully exploited to enhance the model's overall performance in the sequence modeling process.

**Information Extraction.** The information extraction phase is applied to the high-dimensional representations derived from the sequence aggregation stage. Its objective is to distill more valuable sequential information for subsequent tasks while decreasing the computational complexity. For this purpose, We employ a simple and efficient MLP for the information extraction process.

$$R^j = MLP(C^j) \tag{4}$$

The high-dimensional representations are compressed into a lower-dimensional space to form the sequential information, thereby reducing the number of parameters. Note that we use an MLP just because its simplicity, however, it could be replaced by more sophisticated methods if desired.

**Representation Fusion.** The extracted sequential information is concatenated with the original patch embeddings or the outputs from the previous encoder, as illustrated in Figure 2. This process

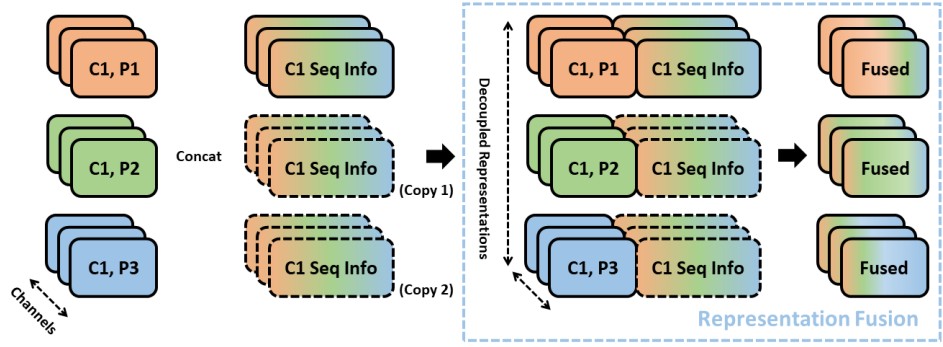

Figure 2: DROSIA encoder concatenates the patch embeddings and extracts sequential information from them. This information is duplicated and combined with the original patch embeddings to create decoupled representations, which are then fused back to the original dimensionality.

resembles residual connection He et al. (2016), but in a decoupled manner, which enhances the information representation capability and improves efficiency, while also facilitating full interaction between the two types of information and optimization for deeper network. Subsequently, the fused representations of patches and sequential information are processed through a normalization layer, where both parts undergo a unified normalization operation. The function is outlined as follows.

$$D = LayerNorm(S^j \circ R^j) \tag{5}$$

$$LayerNorm(H) = \frac{h_i - Mean(H)}{\sqrt{Var(H)}} \tag{6}$$

In Equation (5) and (6), $LayerNorm$ refers to the normalization operation. The $H$ represents the input, while $h_i$ denotes the $i$-th item of $H$. $Mean$ and $Var$ are functions to compute the mean and the variance respectively. Normalization (Kim et al., 2021) aids in optimizing training phase and mitigates the adverse effects of non-stationary processes, which are common in time series data.

Unlike conventional sequence modeling approaches, DROSIA extracts sequential information once per encoder layer, and then aggregates it with the original patch embeddings in a decoupled fashion. For instance, self-attention mechanisms attend to all time points and aggregates sequential information through a weighted sum of points' representations, potentially overlooking individual information and lacks efficiency. In contrast, DROSIA considers sequential information as additional representation and decouples the two, thereby preserving the benefits of both sequential and individual information while circumventing issues such as the quadratic computational complexity.

$$S^{j+1} = FFN(D) \tag{7}$$

Ultimately, we utilize a feed-forward network to facilitate complete interaction between the two types of information, and compress the fused representation to the dimensionality of the input data.

### 3.3 LINEAR DECODING

Once the data has passed through $l$ layers of DROSIA encoder, the output from the final layer, denoted as $s^{l+1}$, is then fed into the linear decoding module to yield the final forecasting results.

$$\hat{Y} = Projection(S^{l+1}) \tag{8}$$

In Equation (8), the $Projection$ is performed using a fully connected linear layer. During the training phase, the model's prediction results are compared against the actual subsequent time series data to compute the error. Subsequently, the parameters of DROSIA are updated using the backpropagation algorithm. The error is quantified using the mean squared error (MSE). Our configuration of the decoding module aligns with numerous previous studies, including PatchTST and iTransformer.

## 4 EXPERIMENTS

**Datasets.** We comprehensively assessed the performance of the DROSIA model on eight multi-variate long-term time series forecasting datasets: Electricity (ECL), four subsets of ETT (ETTm1, ETTm2, ETTh1, and ETTh2), Traffic, Exchange, and Weather. These datasets are publicly available on GitHub[1]. The data processing and split ratio were consistent with TimesNet(Wu et al., 2023).

**Baselines.** We selected several previous state-of-the-art models to conduct extensive experiments, including Transformer-based, such as iTransformer(Liu et al., 2024), PatchTST(Nie et al., 2023), and FEDformer(Zhou et al., 2022), CNN-based, TimesNet(Wu et al., 2023), and MLP-based fore-casters, TiDE(Das et al., 2023), DLinear(Zeng et al., 2023), and FreTS (Yi et al., 2023).

**Settings.** By default, we configure all Transformer-based models with dropout probability $p = 0.1$ and the number of attention heads $n = 16$. For PatchTST and DROSIA, the patch size is 16 with a stride as 8, in line with previous research. When conducting experiments on ECL, and Traffic, both DROSIA and Transformer-based models are equipped with 3 encoder layers, and latent dimension $d = 512$. For smaller datasets, such as Weather, Exchange and ETT subsets, we employ a smaller model size to mitigate the risk of overfitting: 2 layers and $d = 256$. The dimension ratio of the two types of representations (individual versus sequential) within DROSIA is $1 : 1$ across all scenarios.

Table 1: Overall experimental outcomes for long-term time series forecasting, using four prediction horizons: $H \in \{96, 192, 336, 720\}$ across all datasets, and the length of the input $L = 96$, which are consistent with iTransformer (Liu et al., 2024). The results are averaged from these four horizons.

| Category | Ours | Transformer-based | | | CNN-based | MLP-based | | |
|---|---|---|---|---|---|---|---|---|
| Model | DROSIA | iTransformer | PatchTST | FEDformer | TimesNet | TiDE | DLinear | FreTS |
| ETTh1 | **0.441 ± .002** | 0.454 ± .001 | 0.448 ± .003 | 0.453 ± .001 | 0.495 ± .002 | 0.491 ± .000 | 0.465 ± .003 | 0.483 ± .001 |
| ETTh2 | **0.379 ± .002** | 0.389 ± .003 | 0.382 ± .002 | 0.428 ± .001 | 0.417 ± .005 | 0.401 ± .000 | 0.563 ± .001 | 0.531 ± .025 |
| ETTm1 | **0.384 ± .001** | 0.408 ± .001 | 0.388 ± .002 | 0.449 ± .003 | 0.432 ± .002 | 0.424 ± .000 | 0.403 ± .002 | 0.408 ± .001 |
| ETTm2 | **0.281 ± .001** | 0.291 ± .001 | 0.287 ± .002 | 0.301 ± .002 | 0.309 ± .003 | 0.291 ± .001 | 0.349 ± .005 | 0.334 ± .004 |
| Exchange | 0.362 ± .000 | 0.373 ± .006 | 0.368 ± .005 | 0.509 ± .002 | 0.413 ± .002 | 0.367 ± .003 | **0.347 ± .003** | 0.412 ± .009 |
| Weather | **0.255 ± .000** | 0.260 ± .001 | 0.257 ± .001 | 0.302 ± .001 | 0.262 ± .002 | 0.272 ± .001 | 0.265 ± .001 | **0.255 ± .001** |
| ECL | 0.190 ± .001 | **0.185 ± .001** | 0.196 ± .000 | 0.216 ± .001 | 0.192 ± .001 | 0.257 ± .001 | 0.215 ± .001 | 0.202 ± .001 |
| Traffic | 0.479 ± .000 | **0.467 ± .000** | 0.486 ± .000 | 0.621 ± .001 | 0.628 ± .001 | 0.758 ± .002 | 0.643 ± .000 | 0.579 ± .001 |

To reduce the impact of randomness and demonstrate the significance. The outcomes from different prediction horizons are averaged for each dataset, and each experiment is conducted five times to calculate average result and standard deviation, using the mean squared error (MSE) as evaluation metrics. All experiments are performed on a single NVIDIA 4090 GPU with 24GB of memory.

Table 2: Experiments on ECL and Traffic for a fair comparison between DROSIA and iTransformer, involving various lengths of input time series $L \in \{96, 192, 336, 512\}$, and different output horizons: $H \in \{96, 192, 336, 720\}$. Results in **bolded red** indicate the winner in each scenario.

| Length | | 512 | | 336 | | 192 | | 96 | |
|---|---|---|---|---|---|---|---|---|---|
| Model | | DROSIA | iTransformer | DROSIA | iTransformer | DROSIA | iTransformer | DROSIA | iTransformer |
| ECL | 96 | **0.131 ± .001** | 0.135 ± .000 | **0.134 ± .000** | 0.136 ± .001 | **0.141 ± .001** | **0.141 ± .000** | 0.167 ± .001 | **0.158 ± .000** |
| | 192 | **0.150 ± .000** | 0.154 ± .000 | **0.151 ± .001** | 0.155 ± .000 | **0.158 ± .000** | 0.159 ± .001 | 0.176 ± .000 | **0.170 ± .000** |
| | 336 | **0.167 ± .000** | 0.170 ± .000 | **0.170 ± .000** | 0.173 ± .001 | **0.176 ± .001** | 0.177 ± .001 | 0.193 ± .001 | **0.187 ± .001** |
| | 720 | **0.203 ± .000** | 0.206 ± .001 | **0.208 ± .000** | 0.210 ± .000 | **0.215 ± .000** | 0.216 ± .000 | 0.232 ± .000 | **0.224 ± .001** |
| Traffic | 96 | **0.371 ± .000** | 0.395 ± .001 | **0.381 ± .000** | 0.401 ± .001 | **0.401 ± .001** | 0.421 ± .000 | 0.454 ± .001 | **0.434 ± .000** |
| | 192 | **0.389 ± .001** | 0.415 ± .000 | **0.402 ± .000** | 0.423 ± .001 | **0.422 ± .001** | 0.443 ± .000 | 0.466 ± .000 | **0.454 ± .000** |
| | 336 | **0.400 ± .001** | 0.430 ± .000 | **0.419 ± .000** | 0.441 ± .001 | **0.438 ± .001** | 0.459 ± .000 | 0.483 ± .000 | **0.472 ± .001** |
| | 720 | **0.436 ± .000** | 0.472 ± .001 | **0.446 ± .000** | 0.476 ± .000 | **0.466 ± .000** | 0.489 ± .000 | 0.515 ± .001 | **0.507 ± .000** |

### 4.1 EXPERIMENTAL RESULTS

The overall experimental results are presented in Table 1. The **bolded** values denote the best on each dataset, while the underlined indicate the second-best. As observed, DROSIA achieves superior or competitive results compared with other forecasters in all scenarios. However, on large scale datasets, such as ECL (321 channels) and Traffic (862 channels), the channel-independent DROSIA does not outperform the channel-dependent iTransformer model. This comparison is not entirely equitable to DROSIA, as the number of channels significantly exceeds the length of the input data.

---

[1] https://github.com/thuml/Time-Series-Library

Consequently, we adjust the input lengths for ECL and Traffic datasets to facilitate fairer comparisons between DROSIA and iTransformer. Three trails are conducted for this and ablation studies to calculate average MSEs and standard deviations for each scenario. As shown in Table 2, when longer input lengths ($L \geq 192$) are provided, DROSIA could consistently outperform iTransformer on ECL and Traffic, without utilizing any inter-channel information. This outcome underscores the powerful capability of DROSIA in time series modeling and capturing long-distance dependencies.

As shown in Table 1, DROSIA significantly outperforms the MLP-based methods, TiDE, DLinear, and FreTS, across most scenarios. For the Exchange dataset, which comprises 8 channels and is subject to a high degree of randomness and non-stationary, DROSIA still achieves the second-best in MSE, and is only slightly worse than DLinear. In comparison to Transformer-based and CNN-based models, DROSIA consistently exceeds the performance of FEDformer, TimesNet, and PatchTST, and demonstrates superior behavior to iTransformer in datasets with small amount of channels.

Table 3: Efficiency comparisons between DROSIA and various typical time series forecasters with the computational complexity, which is consistent with (Han et al., 2024). DROSIA is the only method that is linear to the input length $L$, prediction horizon $H$, and number of channels $C$.

|  | DROSIA | iTransformer | PatchTST | Transformer |
|---|---|---|---|---|
| Complexity | $O(CL + CH)$ | $O(CL + C^2 + CH)$ | $O(CL^2 + CH)$ | $O(CL + L^2 + HL + CH)$ |

## 4.2 ABLATION STUDY

**Efficiency Analysis.** We assessed the efficiency of DROSIA against various typical forecasters. DROSIA mainly comprises patch embedding, information extraction, representation fusion, and linear decoding modules. Assuming an input length $L$, a number of channels $C$, a patch size $p$ with a stride $s$, model dimension is $d$ with ratio of two types of information $1:1$ ($d/2$ for each), and prediction horizon is $H$. The computational complexities are $O(CpdL/2s)$, $O(Cd^2L/4s)$, $O(Cd^2L/2s)$, and $O(CHd/2)$ respectively. By ignoring all constants, we derive the overall computational complexity of DROSIA as $O(CL + CH)$, which is linear to $L$, $C$, and $H$. The complexity of other models was also computed in this way, as presented in Table 3. DROSIA stands out as the only method with linear complexity to $L$ and $C$ of time series data, demonstrating its high efficiency for time series forecasting tasks, particularly in scenarios of large variate sizes and long input lengths.

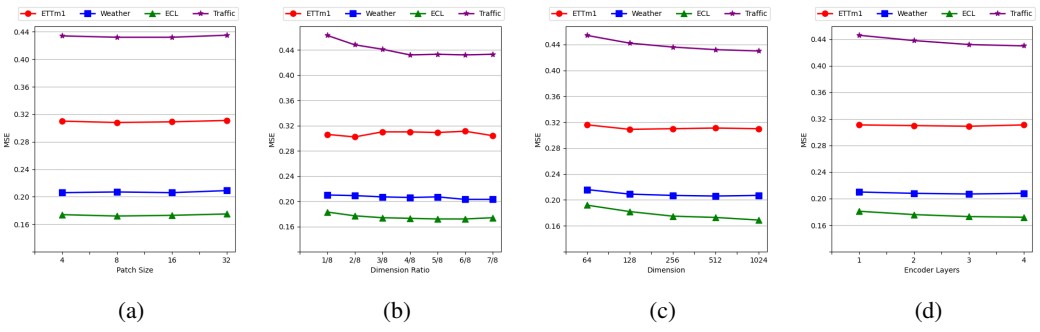

|  | (a) | (b) | (c) | (d) |

Figure 3: Hyperparameter sensitivity analysis of DROSIA. Four datasets with different variate size are adopted: ETTm1, Weather, ECL and Traffic, with the patch sizes: $p \in \{4, 8, 16, 32\}$, dimension ratio of information within each time patch: $r \in \{1/8, 2/8, 3/8, 4/8, 5/8, 6/8, 7/8\}$, model dimension: $d \in \{64, 128, 256, 512, 1024\}$, and number of encoder layers: $n \in \{1, 2, 3, 4\}$.

**Hyperparameter Sensitivity Analysis.** We selected four datasets with varying numbers of channels: ETTm1 (7 channels), Weather (21 channels), ECL (321 channels), and Traffic (862 channels), and conducted a sensitivity analysis on several key hyperparameters of DROSIA, which include the patch size $p$, the dimension ratio of patch embeddings $r$, the model dimension: $d$, and the number of encoder layers $n$. To ensure fairness and avoid bias due to an excessively large patch size, we set the input length to 192. For all scenarios, we used Mean Squared Error (MSE) as the evaluation metric. All other settings were aligned with the default experimental configurations.

As depicted in Figure 3, it reveals that variations in patch size have a negligible impact on the overall performance of DROSIA across all datasets. For the dimension ratio, model dimension, and number of encoder layers, datasets with a large number of variates, such as ECL and Traffic, benefit from increased values of $r$, $d$ and $n$ to achieve improved prediction performance. Conversely, for smaller scale datasets like ETTm1 and Weather, DROSIA does not derive significant advantages from larger values of these hyperparameters, in some cases, the performance even deteriorates.

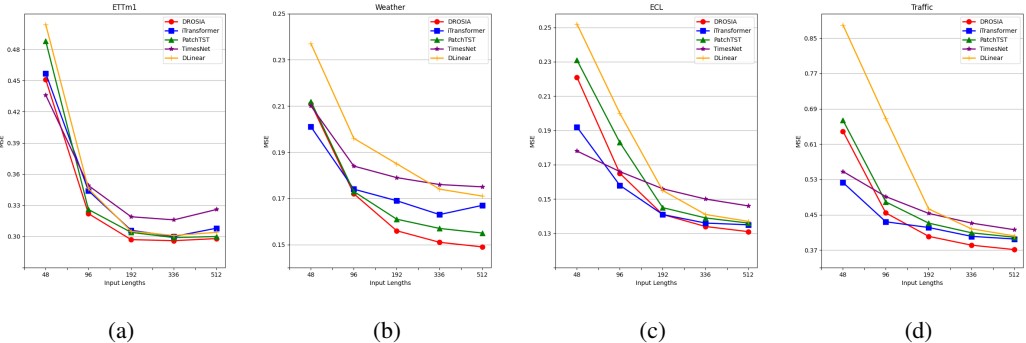

Figure 4: Ablation study of prediction performance of DROSIA, iTransformer, PatchTST, DLinear, and TimesNet. Four datasets with different variate sizes are adopted: ETTm1, ECL and Traffic, with varying input lengths: $L \in \{48, 96, 192, 336, 512\}$, and the prediction horizon $H = 96$.

**Influence of Input Length.** We selected four datasets with varying variate sizes: ETTm1, Weather, ECL, and Traffic, to conduct a detailed analysis on the impact of input length. For comparison, we adopted four baselines, which include Transformer-based models iTransformer and PatchTST, CNN-based model TimesNet, and MLP-based model DLinear. It should be noted that iTransformer and TimesNet are channel-dependent models, whereas the others are channel-independent.

As depicted in Figure 4, the DROSIA model demonstrates its superior effectiveness across all scenarios when compared to channel-independent models like PatchTST and DLinear. It achieves the best performance on all datasets with longer input time series data lengths ($L \geq 192$), even outperforming channel-dependent models such as TimesNet and iTransformer. For datasets with larger variate sizes like ECL and Traffic, TimesNet and iTransformer exhibit superior performance when the input length is set to $48$. However, their advantage diminishes and is eventually overtaken as the input length increases. This trend suggests the value of inter-channel information in time series forecasting and highlights a limitation of channel-dependent models in capturing long-distance dependencies. The question of how to better balance sequential and inter-channel information warrants further investigation. Moreover, the performance of DROSIA is consistent and progressively improves with increasing input length, ultimately achieving state-of-the-art forecasting accuracy. This trend already attests to the model's robust capability in sequence modeling.

**Effectiveness of DROSIA.** We investigate the role that decoupled representations play in the overall performance of DROSIA and the efficacy in time series forecasting. All of the eight datasets are selected for comparison, with the input length $L = 96$, and the prediction horizons $H \in \{96, 192, 336, 720\}$. PatchTST was chosen as the benchmark, which employs the patch embedding and Transformer encoder to integrate sequential information. As indicated in Table 4, When using DROSIA architecture ("P+S"), the performance of both models could significantly surpass that of the setting where only sequential information is utilized across most of the scenarios. While the original PatchTST ("S") performs even worse to "P" on small scale datasets, where patch embeddings are directly fed into the FFN layer. This finding validates the effectiveness of DROSIA in aggregating sequential information and significantly enhances the performance.

**Different Information Extraction Methods.** As discussed in Section 3.2, we utilize an MLP for sequential information extraction primarily due to its simplicity, however, it could be substituted with any methods. To evaluate the impact of various information extractors on the overall effectiveness of DROSIA, we compare five methods: MLP, Self-Attention, CNN, RNN, and Max Pooling. CNN refers to a single convolutional layer, and RNN denotes the vanilla version in this context. The outcomes of these experiments are presented in Table 5.

Table 4: Experiments on the decoupled representations, which encompass three cases: "P+S" indicates the inclusion of both patch and sequential representations, while "P" or "S" signifies only one respectively. The "P+S" configuration of PatchTST means the patch representations and sequential information extracted via Self-Attention are concatenated, whereas "S" refers to the model's original settings. To mitigate the randomness in the results, we utilized two large datasets (ECL and Traffic).

| Model | DROSIA | | PatchTST | | P |
|---|---|---|---|---|---|
| | P+S | S | P+S | S | |
| ETTh1 | **0.441 ± .002** | 0.452 ± .001 | 0.452 ± .002 | **0.448 ± .003** | 0.449 ± .002 |
| ETTh2 | **0.379 ± .002** | 0.385 ± .003 | **0.379 ± .002** | 0.382 ± .002 | 0.382 ± .002 |
| ETTm1 | **0.384 ± .001** | 0.391 ± .001 | **0.385 ± .001** | 0.388 ± .002 | 0.386 ± .001 |
| ETTm2 | **0.281 ± .001** | 0.288 ± .001 | **0.282 ± .001** | 0.287 ± .002 | 0.284 ± .001 |
| Exchange | **0.362 ± .000** | 0.367 ± .006 | **0.355 ± .002** | 0.368 ± .005 | 0.363 ± .005 |
| Weather | **0.255 ± .000** | 0.261 ± .001 | **0.257 ± .000** | 0.257 ± .001 | 0.265 ± .002 |
| ECL | **0.190 ± .001** | 0.201 ± .001 | **0.192 ± .000** | 0.196 ± .000 | 0.203 ± .001 |
| Traffic | **0.479 ± .000** | 0.496 ± .000 | **0.483 ± .000** | 0.486 ± .000 | 0.559 ± .001 |

Table 5: Experiments for the analysis to five different sequential information extraction methods: MLP (ours), Self-Attention, CNN, RNN, and Max Pooling of DROSIA, with the prediction horizon $H \in \{96, 192, 336, 720\}$, and the length of input time series data $L = 96$ for all of the 8 datasets. All of the results are averaged from four horizons. The **bolded** values denote the best performance, and the underlined values denote the second-best, which are consistent with Table 1.

| Method | MLP | Self-Attention | CNN | RNN | Max Pooling |
|---|---|---|---|---|---|
| ETTh1 | **0.441 ± .002** | 0.452 ± .002 | 0.453 ± .006 | 0.445 ± .003 | 0.445 ± .001 |
| ETTh2 | 0.379 ± .002 | 0.379 ± .002 | 0.380 ± .002 | **0.378 ± .001** | **0.378 ± .002** |
| ETTm1 | **0.384 ± .001** | 0.385 ± .001 | 0.387 ± .002 | 0.387 ± .001 | 0.387 ± .002 |
| ETTm2 | 0.281 ± .001 | 0.282 ± .001 | **0.278 ± .001** | **0.278 ± .000** | 0.280 ± .001 |
| Exchange | 0.362 ± .000 | **0.355 ± .002** | 0.365 ± .003 | 0.357 ± .002 | 0.360 ± .005 |
| Weather | **0.255 ± .000** | 0.257 ± .000 | 0.261 ± .000 | 0.260 ± .001 | 0.264 ± .000 |
| ECL | **0.190 ± .001** | 0.192 ± .000 | 0.195 ± .002 | 0.200 ± .001 | 0.199 ± .000 |
| Traffic | **0.479 ± .000** | 0.483 ± .000 | 0.485 ± .001 | 0.482 ± .001 | 0.491 ± .000 |

The results indicate that each sequential information extraction method could excel on different datasets with smaller variate sizes, such as the four subsets of ETT, Exchange, and Weather, and there are no big gaps in the results of each method. However, for datasets with larger variate sizes, such as ECL and Traffic, the MLP-based method proves to be more effective. These findings suggest that for smaller datasets, there is minimal distinction between various extractors, which underscores the universal effectiveness of DROSIA and the inherent data variability across these datasets. In contrast, more complex datasets necessitate more advanced sequential information extraction methods to achieve optimal performance.

## 5 CONCLUSION AND FUTURE WORK

This paper introduces a novel approach - DROSIA, which incorporates rich temporal relationships as additional representations within each time patch. This method achieves sequential information aggregation in a decoupled fashion, effectively balancing sequential and individual information with linear complexity for sequence modeling. Through comprehensive experimentation, we show that DROSIA attains state-of-the-art performance, particularly in scenarios involving long sequences and large scale data. Compared with previous top-performing channel-dependent models, the channel-independent DROSIA exhibits superior performance across all datasets with a fair the input sequence length when compared with the amount of channnels. Notably, DROSIA does not rely on inter-channel information, highlighting its efficacy in sequence modeling and capturing long-distance dependencies. We contend that DROSIA is broadly applicable to a variety of scenarios.

In the ablation study, we have thoroughly demonstrated the efficacy of DROSIA through a multitude of meticulously designed experiments. However, we also observed that when the input length of time

series is inadequate and the dataset has a large variate size, the prediction accuracy of DROSIA may fall short of channel-dependent methods. This underscores the significance of inter-channel information. Consequently, our future research will concentrate on integrating inter-channel information without excessively compromising the information within each channel, while also considering the model's overall efficiency to achieve a better balance. Through additional experiments (not detailed in this paper), we have verified that inter-channel information significantly diverges from sequential information, necessitating distinct integration strategies. Simply applying DROSIA to inter-channel information aggregation may not be feasible. Overall, this paper presents a successful method for enhanced intra-channel modeling and identifies a challenging research direction in time series forecasting: how to efficiently model both intra- and inter-channel information simultaneously.

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
