# OpenReview forum: "DROSIA: Decoupled Representation on Sequential Information Aggregation for Time Series Forecasting"
_ICLR.cc/2025/Conference — ICLR 2025 Conference Withdrawn Submission_

### Official Review · Reviewer_Fnug · 2024-10-27

**Soundness:** 3
**Presentation:** 2
**Contribution:** 2
**Rating:** 6
**Confidence:** 3

**Summary:**

Authors proposed DROSIA a method of time-series forecasting that incorporates both point-wise and temporal information by applying the following steps:
- Patching similar to existing methods e.g., PatchTST
- DROSIA encoding
    - Sequence aggregation I.e., applying multiple encoding layers on vectorized patches
    - Information extraction I.e., a MLP on concatenated representation of the previous step
    - Representation fusion i.e., layer normalization of concatenated representations of information extraction + vectorized patches followed by a fully connected transformation (similar to residual connections)
- Decoding (projection) i.e., making predictions

Authors studied performance of DROSIA on state-of-the-art of time series forecasting benchmarks and compared with some of the well-known methods in this area.

**Strengths:**

- Authors have done extensive experiments ranging from benchmarks to complexity
- DROSIA achieved better performance with relatively simpler model in Long-term forecasting task

**Weaknesses:**

Although average over 3 trials is reported, standard deviations are not reported. Reporting standard deviation is crucial when performance gap is small. Particularly, when authors claim “significantly outperforming” a method this needs to be confirmed by conducting a statistical test e.g., t-test or Wilcoxon test (based on assumptions).

The proposed method needs adjustment in input length to outperform iTransformer particularly on datasets with a lot of variables and shorter horizons e.g., 96 which could be a disadvantage in applicability of the proposed method on real-world applications. Authors did not provide any instruction on how to find “sufficiently long” input length for their method.

Recently LLM-based methods for time series forecasting have shown state-of-the-art performance [1-4] some of which also based on patching [2] but there is no indication of this category in neither related works nor compared methods. Examples are:
Just to be clear, I am not asking authors to compare with all of these LLM-based methods but I’d like to know their at least their thoughts on positioning this line of work in their study.

Authors have compared their method with DLinear which is based on MLP that utilizes point-wise information but there are also MLP-based models such as [5] that incorporate global and local information which to me are more similar to the proposed method but is missing from compared methods.

One missing ablation experiment is related to the equation (7). What is the performance without this component?

Some reproducibility information is missing such as code (cited in the abstract that it will be provided later), learning rate or any utilized regularizations.

Original (non-averaged) results of table 1 should be provided in the appendix


Smaller fixes:

- Typo on page 4 “In Equation (4)” -> Equation (3)

- Typo on page 5 “In Equation (3) -> Equation (8)

- Font size in figures is too small (at least for me)

- It would be helpful to add another row “average+- standard deviation” to each table to summarize the results per analysis

References

[1] Large language models are zero-shot time series forecasters. Neurips, 2023.

[2] One Fits All: Power General Time Series Analysis by Pretrained LM. Neurips, 2023.

[3] Time-LLM: Time Series Forecasting by Reprogramming Large Language Models. ICLR, 2024.

[4] TEST: Text Prototype Aligned Embedding To Activate LLM’S Ability for Time Series. ICLR, 2024.

[5] Frequency-domain MLPs are More Effective Learners in Time Series Forecasting. Neurips, 2023.

**Questions:**

Do you have any intuition for certain behaviours in your sensitivity analysis? e.g., why patch size is not important, why for larger dataset in terms of number of channels e.g., Traffic high values of almost all hyper-parameters makes things worse?

Is there any parameter sharing in DROSIA?

Are observations made in table 5 going to hold for longer horizons e.g., H=720?

What is “P” model in table 4 last column?

Compared baselines are also applicable to other time-series tasks including classification, short-term forecasting, imputation, and anomaly detection and often competitive in said tasks, but DROSIA is only studied in long-term forecasting do you have any sense on applicability of DROSIA in other time series tasks?

I would be happy to revise my score if authors clarify questions/weaknesses particularly:
- Missing experiments e.g., ablation or potentially missing baselines as well as information to evaluate performance better e.g., standard deviation or significance test.
- Issue with number of features/channels and input length

Update: I'd like to thank the authors for engaging during the rebuttal period to address reviewer's comments. After reading their responses and comments from other reviews, I have decided to increase my score.

---

> ### Author Response · Authors · 2024-11-13
>
> Thank you for the comments and suggestions. We have carefully considered each point and detail our responses below:
>
> For weaknesses:
>
> We apologize for the inaccurate use of the term “significantly” while only reporting the average results of three trials, and will conduct a statistical test as soon as possible, report the results here and in the paper.
>
> We believe that there is no perfect model, but only applicable scenarios for the model. We have demonstrated the effectiveness and efficiency of DROSIA in its applicable scenarios and consider that the requirements for these scenarios are not difficult to meet.
>
> * For example, the Weather dataset includes 21 indicators, such as temperature and humidity, recorded every 10 minutes. So we can wait 1 day and 8 hours to obtain a sequence of length 192, but we need 21 different types of sensors to gather the 21 channels. Hard to imagine what express delivery could bring 21 (or even 192) types of sensors home and have them functioning properly within 1 day and 8 hours, lol. Increasing the sequence length is not necessarily more difficult than increasing the number of channels.
>
> * Moreover, we can adjust to a finer time granularity or quickly obtain longer sequences through methods like interpolation to address this issue.
>
> * We have honestly reported the shortcoming of DROSIA in experiments and affirm the value of inter-channel information, emphasizing the importance of efficiently modeling both intra- and inter-channel information in future work.
>
> * DROSIA is slightly weaker than channel-dependent models only in the scenario of extremely many channels and short sequences, which might sound quite rare when described. We believe that this shortcoming of DROSIA cannot overshadow its success in other, more extensive fields.
>
> We are very willing to position LLM-based forecasters in our study if you insist, and please allow me to explain why we haven’t done so before.
>
> * Paper [1] (https://arxiv.org/pdf/2406.16964) thoroughly analyzed the LLM-based methods in the field of time series forecasting, and they claimed: "Popular LLM-based time series forecasters perform the same or worse than basic LLM-free ablations, yet require orders of magnitude more compute", and “Despite the recent popularity of LLMs in time series forecasting, they do not appear to meaningfully improve performance”.
>
> * Our motivation of DROSIA is to demonstrate its effectiveness and efficient in sequence modeling and time series forecasting, so we did not compare it with LLM-based forecasters but with various transformer-based models.
>
> * Research related to LLM requires many resources, which can be quite challenging for ordinary researchers (like me). Therefore, I hope to focus on small but meaningful work.
>
> We will compare our model with FreTS you mentioned and other potential baselines as soon as possible, report the results here and in the paper.
>
> We are pleased to conduct an ablation study for this component. However, the equation (7) in our paper fuse two types of information and compress the fused representation to the original dimension. It enables a full interaction between the two, and could not be simply removed in scenarios of multi-layer encoder. How would you like to conduct this ablation study?
>
> We have uploaded our codes and scripts to the supplementary material, which could be directly added to the Time-Series-Library (TSL) repo in GitHub (https://github.com/thuml/Time-Series-Library), and follow their running instructions. Our github link will be added in the paper after the anonymous period ends.
>
> We would add the non-averaged results of table 1 in the appendix, and fix typos, font size, and add another row “average+- standard deviation” to each table as soon as possible.
>
> For questions:
>
> Because of the limit of characters, we would add another official comment for responses to the questions.
>
> References
>
> [1] Are Language Models Actually Useful for Time Series Forecasting? Neurips, 2024

---

> > ### Comment · Reviewer_Fnug · 2024-11-19
> >
> > Thank you for your response.
> >
> > - Re ablation on Equation 7, your clarification on Scenario "P" (in the following comment) addressed my concern to some extent and no action is needed on your end for this.
> >
> > - Re LLM, thank you for sharing "Are Language Models Actually Useful for Time Series Forecasting? Neurips, 2024" seems like an interesting read. I think an overall discussion of LLM-based methods in the related work section can be beneficial to readers. I recognize that running LLM related experiments "can" be time consuming but nor for methods like "Large language models are zero-shot time series forecasters. Neurips, 2023" where the LLM component is frozen. The reason that this comparison could be interesting is that when looking at Table 9 of the paper you shared it seems LLM-based methods are doing better compared to the no-LLM baseline on longer windows and larger datasets setups which partially aligns with your claim.
> >
> > - Re input length/features, thank you for providing the intuition but I am not sure if it answers my concern on what is “sufficiently long” input length.

---

> ### Author Response · Authors · 2024-11-13
>
> For questions:
>
> Firstly, we need to clarify that patch size does indeed have an impact.
>
> * When the input length L=96 and the patch size p=32, the model’s performance declines. This is why we set the input length to 192 when analyzing the impact of various patch sizes, to enable a fairer comparison.
>
> * We believe that the model’s predictive performance is related to the number of parameters and the depth of the network. Larger datasets require more parameters and deeper networks. When the patch size is set close to the input length, the sequence of patches is too short, leading to an insufficient number of parameters for the sequence information extractor, which is also evident from the analysis of the other three hyperparameters.
>
> * Since the sequence information is processed by the extractor from the concatenated patch embeddings, when the dimension ratio of patch embeddings is sufficiently high, the number of extractor parameters is also sufficient. The analysis of larger dimensions could confirm this point as well, and the number of encoding layers could confirm the impact of network depth on predictive performance.
>
> There is no parameter sharing in DROSIA.
>
> We have tested the longer horizons in Figure 4, but not in table 5, and we are willing to do so.
>
> Scenario "P" means only original patch embeddings that are directly fed into FFN layer (equation 7), the dimension of patch embeddings is set to d for this, while d/2 for "P+S".
>
> We are very willing to evaluate the effectiveness of DROSIA in other time series tasks, but in this paper, we focused on forecasting to demonstrate the capabilities of sequence modeling and capturing long-distance dependencies of DROSIA.
>
> Thank you for your open-mindedness towards DROSIA. We will take your suggestions seriously and make revisions as soon as possible, hoping to gain your approval in the end.

---

> ### Author Response · Authors · 2024-11-16
>
> We apologize for neglecting a point in the previous response to question 1: "why for larger dataset in terms of number of channels e.g., Traffic high values of almost all hyper-parameters makes things worse".
>
> * We need to clarify that in the hyperparameter sensitivity analysis, since we use MSE to evaluate the model's performance, as the hyperparameter values increase, the MSE of DROSIA on Traffic decreases, which indicates that the predictive performance is better, not worse.
>
> * This is consistent with the viewpoint expressed in our previous response, which states that larger datasets require more parameters and deeper networks.
>
> Before we add the baseline and statistical analysis, we would like to explain the two issues you are particularly concerned about, and why we did not do these.
>
> * Our motivation is to demonstrate the effectiveness and efficiency of DROSIA, so we choose many previous best-performing models: PatchTST, TiDE, TimesNet, DLinear, and FEDformer with different sequence modeling methods, and choose iTransformer as previous state-of-the-art channel-dependent forecasters. We thought these models are sufficient for our motivation before. However, the FreTS is indeed more similar to DROSIA, which we think should be added as a baseline in our paper, and we are working on it.
>
> * To our knowledge, the standard deviation or significance test is often conducted for the probabilistic time series forecasting methods, such as the VAE-based or diffusion-based forecasters, not in the deterministic modeling. Our experimental settings (average over 3 trials) are consistent with iTransformer, as well as other deterministic forecasters.
>
> * The only difference between DROSIA and PatchTST is in the encoder, so we think, although DROSIA surpasses PatchTST with small gaps on simple datasets, since 1. All hyperparameters are the same, we are not better tuning DROSIA over PatchTST; 2. There are bigger gap on larger datasets (table 1). 3. As the input length increases, DROSIA surpasses PatchTST at a greater margin (figure 4). 4. The performance of PatchTST could benefit from DROSIA architecture (table 4). We think the existing results in our paper are sufficient to demonstrate the effectiveness of DROSIA over Transformer (or PatchTST), especially for larger datasets and long input lengths.
>
> * However, we are very willing to add standard deviation or significance test results, and we are working on it.
>
> We will add FreTS as baseline, and conduct a standard deviation for all tables, and report the results soon. Is there any other concern of you, please tell us :)
>
> For issue with number of features/channels and input length, is there still any concern of you, please tell us :)

---

> > ### Comment · Reviewer_Fnug · 2024-11-23
> > **Clarification on standard deviation**
> >
> > If all the utilized methods are fully deterministic, what are the standard deviation numbers that you are reporting for the new experiments?

---

> ### Author Response · Authors · 2024-11-17
>
> We have conducted 5 trails of DROSIA, iTransformer, PatchTST, and FreTS on 4 ETT subsets with the newest version of TSL repo. All the parameters are used just the same as the scripts we uploaded in supplementary material for all forecasters. The MSE results are reported here, and the same phenomenon is also found for the MAE metric.
>
> We will revise our paper to add these results with our analysis, and we are still working for larger datasets and tables in the ablation study. We could finally demonstrate the significance of DROSIA over these baselines on small datasets, as the input length increases, DROSIA could surpass all baselines with even greater margin, which should be able to demonstrate the effectiveness of DROSIA in extensive scenarios.
>
> |Model|DROSIA|iTransformer|PatchTST|FreTS|
> |:-:|:-:|:-:|:-:|:-:|
> |ETTh1|0.441 $\pm$ .002|0.454 $\pm$ .001|0.448 $\pm$ .003|0.483 $\pm$ .001|
> |ETTh2|0.379 $\pm$ .002|0.389 $\pm$ .003|0.382 $\pm$ .002|0.531 $\pm$ .025|
> |ETTm1|0.384 $\pm$ .001|0.408 $\pm$ .001|0.388 $\pm$ .002|0.408 $\pm$ .001|
> |ETTm2|0.281 $\pm$ .001|0.291 $\pm$ .001|0.287 $\pm$ .002|0.334 $\pm$ .004|
>
> We will also report the results for larger datasets, other baselines, and ablation studies soon. Thanks for your suggestions, we will keep to conduct standard deviation in all of our future studies, also thanks for your kindness for mention FreTS as the additional baseline, which is similar to DROSIA and is already provided in TSL.

---

> > ### Comment · Reviewer_Fnug · 2024-11-18
> > **Need extra context for the new experimental results**
> >
> > Thanks for your efforts in providing new experimental results and including FreTS which surprisingly performed poorly. I am a bit confused by the new results, you mentioned it is based on the newest version of TSL repo. What has changed compared to the version that you utilized originally? because compared to Table 1, DROSIA is performing worse (higher MSE in ETTh1, m1, and m2) while others e.g., iTransformer is roughly the same. Did you make any other changes compared to the experiments in Table 1?

---

> ### Author Response · Authors · 2024-11-19
> **Extra context for the new experimental results**
>
> Thank you for your response.
>
> We need to explain for this. DROSIA performs roughly the same on ETTh1 and ETTm1, and worse on other 2 datasets. While iTransformer performs worse on ETTh2 and PatchTST performs worse on 3 datasets except ETTm1.
>
> We checked previous python training file and scripts of TSL, and apologize that we only use 1 encoder layer for all forecasters on small datasets before and report 2 in our paper. We have run 5 trials with the new scripts (2 encoder layers on small datasets) for all models, and we have uploaded these scripts in the supplementary material. The new results could demonstrate the effectiveness and significance of DROSIA over other forecasters on small datasets.

---

> ### Author Response · Authors · 2024-11-19
>
> Thank you for your response. Please let me explain for this.
>
> * We are very happy that our responses have addressed your concerns to this issue.
>
> * We will revise our paper to add an overall discussion of LLM-based mthods soon, and we agree that the LLM-based forecasters are indeed important to the research fields of time series analysis. However, we still concern about their "the same or worse" performance and inefficiency, at least for the existing LLM-based forecasters.
>
> * We apologize that our responses did not address your concern for this issue, and we will try our best to demonstrate the importance of DROSIA to sequence modeling and time series forecasting.
>
>   1. The channel-dependent models, such as iTransformer and TimesNet, could perform better on large datasets with short input lengths not because their effective sequence modeling, but the inter-channel information that they integrate into the intra-channel information.
>
>   2. The comparison is not fair for channel-independent models, such as DROSIA. Because they do not utilize any inter-channel information, and the number of channels significantly exceeds the length of the input data in these scenarios, e.g. ECL (321 channels) and Traffic (862 channels) with only input lengths as 48 or 96.
>
>   3. These scenarios are not easy to meet, as we mentioned in our previous responses: Increasing the input length is not necessarily more difficult than increasing the number of channels. Moreover, we can adjust to a finer time granularity or quickly obtain longer sequences through methods like interpolation to address this issue.
>
>   4. As we mentioned in ablation study (figure 4): The advantage of channel-dependent forecasters over DROSIA diminishes and is eventually overtaken as the input length increase (longer than 96). The channel-dependent use both intra- and inter-channel information, while DROSIA only use intra-channel information, and with only linear computational complexity, which demonstrate the effective and efficient sequence modeling ability of DROSIA (our motivation).
>
>   5. DROSIA surpasses all channel-independent forecasters in all scenarios, surpasses all forecasters on small datasets, and could surpass channel-dependent models on large datasets with a fair input length (not too short when compared to the channels). This require is not difficult to meet. With the new experimental results we reported here, DROSIA also significantly surpasses iTransformer, PatchTST, and FreTS on small datasets, and we will report the results on large datasets soon.
>
> All models have their respective limitations, as we have discussed the advantages and disadvantages of the channel-dependent models in the experiment section, they benefit from inter-channel information, which allows them to perform well on datasets of many channels when the input length is short. However, they sacrifice too much temporal information (intra-channel) to maintain high efficiency, leading to a decline in performance on fewer-channel datasets or when the input length is longer. This demonstrates our motivation and aligns with our view that there is no perfect model, only models that are suitable for their respective scenarios. We have truthfully reported the limitations of DROSIA on datasets of many channels with short input lengths, yet the limitations are not difficult to overcome, and DROSIA achieves state-of-the-art in all other scenarios.

---

> ### Author Response · Authors · 2024-11-19
>
> We have run 5 trails for DROSIA, iTransformer, PatchTST, and FreTS on large datasets. The performance of FreTS is very poor in Traffic, as well as other MLP-based models, e.g. DLinear and TiDE (table 1). We think there may be an underfitting phenomenon of these methods in this scenario (input lengths are only set to 96). From figure 4 in our paper, we can see that DLinear performs much better on large datasets (figure 4c and 4d) when the input lengths increase, and could even reach roughly the same as PatchTST and iTransformer (when input length is 512).
>
> |Model|DROSIA|iTransformer|PatchTST|FreTS|
> |:-:|:-:|:-:|:-:|:-:|
> |ECL|0.190 $\pm$ .001|0.185 $\pm$ .001|0.196 $\pm$ .000|0.202 $\pm$ .001|
> |Traffic|0.479 $\pm$ .000|0.467 $\pm$ .000|0.486 $\pm$ .000|0.579 $\pm$ .001|

---

> ### Author Response · Authors · 2024-11-20
>
> We have revised our paper to add discussions about all LLM-based papers that were mentioned in your comments in the related work section. We also fixed the typos and updated the citation information of several newly accepted papers, and we are still working for adding the new experimental results and analysis related to these results.
>
> We found some compilation error for the name of a cited paper, so we recompile our paper and have uploaded it again.

---

> ### Author Response · Authors · 2024-11-21
>
> Additional response to weakness 2.
>
> We reread your concern about the input data length and realized that we may have misunderstood it previously. You would like us to provide methods to find what is “sufficiently long” or how long is enough, rather than the limitations in this scenario. Please let me explain for this.
>
> The record lengths of each dataset we use in our experiments are sufficiently long (at least for thousands). While the number of channels are much less than the data lengths. So we can simply adjust the input length the same as the number of channels for ECL and Traffic for fair comparison.
>
> For various real-world application scenarios, since increasing the sequence length is not more difficult than increasing the number of channels, it is always possible to meet the requirement of having the sequence length consistent with the number of channels.

---

> ### Author Response · Authors · 2024-11-24
> **Our clarification on standard deviation**
>
> We sincerely thank you for the open-mindedness towards our paper, as well as the detailed communication and interaction with us.
>
> Although we focus on the deterministic models, there are still randomnesses in the experimental results, due to the random initialized parameters. We have realized the importance of standard deviations to demonstrate the significance when the gaps are small, even for deterministic models. Thank you for this :)
>
> We have made substantial revisions to our paper, please refer to the general response to all reviewers in the beginning of this web page.

---

> > ### Comment · Reviewer_Fnug · 2024-11-25
> > **Thank you**
> >
> > I'd like to thank authors again for their engagement during the rebuttal period since the beginning of this phase. I have added a note regarding increasing my score to the original review post and after the latest comments I remain in favour of acceptance.

---

> > > ### Author Response · Authors · 2024-11-26
> > > **We sincerely thank you**
> > >
> > > We have learned a lot from the interactions with you, sincerely thank you for this.
> > >
> > > We are very happy to gain your approval in the end.
> > >
> > > Best wishes :)

---

### Official Review · Reviewer_XqSP · 2024-11-03

**Soundness:** 2
**Presentation:** 2
**Contribution:** 2
**Rating:** 5
**Confidence:** 4

**Summary:**

For the task of time series forecasting, authors propose decoupled representation to integrate temporal relationships once as an additional representation for each point, achieving sequential information aggregation in a decoupled fashion.

**Strengths:**

- Nice paper writing.
- Enough ablation study is also included.

**Weaknesses:**

- The idea of decoupled representation lacks novelty. The technical implementation is relatively weak.
- Lack of in-depth analysis, e.g., channel-dependent model.
- No available codes.

**Questions:**

- What is the big difference between your work and PatchTST?
- You consider iTransformer as a strong baseline, but the results in Table 1 seem to be different from the results in iTransformer. Any experimental settings changed?
- Efficiency analysis in Table 3 should include more linear-based methods: DLinear, and TiDE. It would be great to conduct more analysis like Figure 10 in iTransformer rather than only computational complexity.

**Details Of Ethics Concerns:**

Nan

---

> ### Author Response · Authors · 2024-11-13
>
> Thank you for the comments and suggestions. We have carefully considered each point and detail our responses below:
>
> For weaknesses:
>
> * We would like to know more about your concerns of novelty and the technical implementation, could you please clarify that why you think our idea lacks novelty, and where of the technical implementation is weak? This would be important for us to improve our paper.
>
> * Our motivation of this paper is to demonstrate the effectiveness and efficiency of DROSIA for sequence modeling and capturing long-distance dependencies. The reason that we choose the previously state-of-the-art channel-dependent model - iTransformer is also to demonstrate the motivation. Consequently, we believe that we should focus on various intra-channel modeling methods.
>
> * We have uploaded our codes and scripts to the supplementary material, which could be directly added in the Time-Series-Library (TSL) repo in GitHub (https://github.com/thuml/Time-Series-Library), and follow their running instructions. Our github link will be added in the paper after the anonymous period ends.
>
> For questions:
>
> * We replace the Transformer encoder of PatchTST with DROSIA encoder, which is much different when compared to Transformer (please refer to the methodology section and our response to the question 1 regarding to the theory of reviewer eMSN), and we conduct extensive experiments and ablation studies to demonstrate the effectiveness and efficiency of DROSIA for sequence modeling and capturing long-distance dependencies.
>
> * Because we unified the batch size and learning rate for all datasets and forecasters, and please let me explain for this.
>
>   1. We conduct all of the experiments based on the Time-Series-Library (TSL) repo, which is open-sourced by the research team of iTransformer.
>
>   2. TSL provides many scripts for model training. However, we found that the hyperparameters set in the scripts of different models are not always the same, such as the number of encoder layers and attention heads, batch sizes, learning rates, and so on.
>
>   3. We could reproduce the results of iTransformer reported in the corresponding paper using the provided scripts, however, the forecasting performance of other models (including DROSIA) could also benefit from the hyperparameters that are used for iTransformer.
>
>   4. To conduct a fair comparison, we set all hyperparameters to be the same. Not only for the hyperparameters we introduced in our experimental settings, we also set others to the default settings that the TSL team provided in their python model training file, but not to the scripts.
>
> * We believe that there is no perfect model, but only applicable scenarios for the model. DROSIA is not a hexagonal warrior, but it could reach the state-of-the-art performance over all channel-independent models, and over channel-dependent models with a fair input length of time series when compared to the number of channels, with only linear complexity. We think this is enough for us to demonstrate our motivation of this paper.

---

> ### Author Response · Authors · 2024-11-17
>
> For the concerns for the novelty, please refer to our response to the question 1 regarding to the theory of reviewer eMSN.
>
> Is there any concern of you, please tell us :)

---

> ### Author Response · Authors · 2024-11-21
>
> Additional response to weakness 2.
>
> We have discussed the advantages and disadvantages of the channel-dependent models in the related work and experiment sections, they benefit from inter-channel information, which allows them to perform well on datasets of many channels when the input length is short. However, they also have their limitations, as they sacrifice too much temporal information (intra-channel) to maintain high efficiency, leading to a decline in performance on fewer-channel datasets or when the input length is longer. This demonstrates our motivation and aligns with our view that there is no perfect model, only models that are suitable for their respective scenarios. We have truthfully reported the limitations of DROSIA on datasets of many channels with short input lengths, yet it achieves state-of-the-art in all other scenarios.

---

### Official Review · Reviewer_Mmv4 · 2024-11-08

**Soundness:** 2
**Presentation:** 3
**Contribution:** 2
**Rating:** 3
**Confidence:** 4

**Summary:**

This paper presents Decoupled Representation On Sequential Information Aggregation (DROSIA) for time series forecasting. The key idea is to aggregate sequential information in a decoupled fashion, effectively balancing it with individual point information. The experimental results demonstrate the effectiveness of the proposed method.

**Strengths:**

* The paper is well-written and organized.
* The proposed network architecture appears to be novel.
* The method is lightweight, utilizing relatively fewer parameters than existing approaches.

**Weaknesses:**

* There are clarity issues in explaining the proposed method. Some components and their roles are not clearly justified.
* Certain comparisons may not be entirely fair. In addition, several related works are not mentioned or compared (see the questions below).
* For datasets with a small number of variates or low complexity, the improvements over other models, such as PatchTST, are marginal.

**Questions:**

1. It is not clear why each component of the proposed DROSIA is designed in its current form. What advantages do the chosen design choices provide compared to potential alternatives?
2. It may not be fair to compare with channel-dependent methods e.g., iTransformer, as they have less training data.
3. Some important channel-independent and channel-dependent baselines are not mentioned or compared, such as:

[1] Unified Training of Universal Time Series Forecasting Transformers (ICML 2024)

[2] Chronos: Learning the Language of Time Series (arXiv preprint arXiv:2403.07815)

[3] One Fits All: Power General Time Series Analysis by Pretrained LM (NeurIPS 2023)

[4] S²IP-LLM: Semantic Space Informed Prompt Learning with LLM for Time Series Forecasting (ICML 2024)

[5] TimeMixer: Decomposable Multiscale Mixing for Time Series Forecasting (ICLR 2024)

4. What is the performance of DROSIA in few-shot scenarios compared to the baselines?

---

> ### Author Response · Authors · 2024-11-13
>
> Thank you for the comments and suggestions. We have carefully considered each point and detail our responses below:
>
> For weaknesses:
>
> * We apologize for the impression our paper has given you. Could you please specify which components were not clearly justified? This information is of great significance for us to improve the paper.
>
> * Please see our responces to the questions.
>
> * We will conduct a statistical test as soon as possible to demonstrate the significance of DROSIA over PatchTST in small datasets, and report the results here and in the paper for your concerns. Additionally, please allow me to reiterate the motivation of this paper.
>
>   1. We aim to construct a simple, effective and efficient method for sequence modeling and have chosen to demonstrate its effectiveness using a variety of time series forecasting datasets.
>
>   2. PatchTST employs the Transformer encoder for sequence modeling, and the only difference between our approach and theirs is in the encoder. However, our method outperforms PatchTST, especially on complex datasets, as evidenced by Table 1.
>
>   3. Moreover, when longer sequences are input, DROSIA surpasses PatchTST by an even greater margin (figure 4). These results are sufficient to prove the effectiveness of our proposed method in sequence modeling and capturing long-distance dependencies, and it does so with only linear complexity.
>
> For questions:
>
> 1. We want to balance between individual and sequential information, and reduce the computational complexity, which have been explained in the paper. This intuition is similar to the local-global architectures, and we emphasize the importance of local information. Moreover, we conduct extensive ablation studies for the comparison to potential alternatives.
>
> 2. All of the baselines are trained and evaluated with the same split settings of data, as well as the batch size, channels, input length, prediction hirizon, and so on. It is consistent with other work in the field of time series forecasting. We are confused of the "less training data" of channel-dependent models, could you please be more specific of what you mean?
>
> 3. We are very willing to introduce other important forecasters, and may have missed some models and will add them as soon as possible. However, this work focuses on the effectiveness and efficiency of DROSIA for sequence modeling and capturing long-distance dependencies, which has been demonstrated.
>
> 4. We have not evaluated the ability of DROSIA in few-shot scenarios, this is a valuable direction for us. In this paper, we focus on the effectiveness and efficiency of DROSIA for sequence modeling and capturing long-distance dependencies, rather than transfer learning.

---

> ### Author Response · Authors · 2024-11-17
>
> For the weakness 3 you mentioned, we have conducted 5 trails of DROSIA, iTransformer, PatchTST, and FreTS on 4 ETT subsets with the newest version of TSL repo (https://github.com/thuml/Time-Series-Library). All the parameters are used just the same as the scripts we uploaded in supplementary material for all forecasters. The MSE results are reported here, and the same phenomenon is also found for the MAE metric.
>
> With these results, we could demonstrate the significance of DROSIA over these baselines on small datasets, as the input length increases, DROSIA could surpass all baselines with even greater margin, which should be able to demonstrate the effectiveness of DROSIA in extensive scenarios.
>
> |Model|DROSIA|iTransformer|PatchTST|FreTS|
> |:-:|:-:|:-:|:-:|:-:|
> |ETTh1|0.441 $\pm$ .002|0.454 $\pm$ .001|0.448 $\pm$ .003|0.483 $\pm$ .001|
> |ETTh2|0.379 $\pm$ .002|0.389 $\pm$ .003|0.382 $\pm$ .002|0.531 $\pm$ .025|
> |ETTm1|0.384 $\pm$ .001|0.408 $\pm$ .001|0.388 $\pm$ .002|0.408 $\pm$ .001|
> |ETTm2|0.281 $\pm$ .001|0.291 $\pm$ .001|0.287 $\pm$ .002|0.334 $\pm$ .004|

---

> ### Author Response · Authors · 2024-11-20
>
> We have revised our paper to add citations or discussions about paper 3,4,5 that were mentioned in your comments in the related work section, and we have carefully read other 2 papers you mentioned (MOIRAI and Chronos), which are interesting and meaningful studies. However, the 2 methods focus on zero-shot performance of pre-trained time series models, which is much different with our motivation.
>
> Please allow me to explain why we do not compare DROSIA with LLM-based forecasters.
>
>   * Paper [1] (https://arxiv.org/pdf/2406.16964) thoroughly analyzed the LLM-based methods in the field of time series forecasting, and they claimed: "Popular LLM-based time series forecasters perform the same or worse than basic LLM-free ablations, yet require orders of magnitude more compute", and “Despite the recent popularity of LLMs in time series forecasting, they do not appear to meaningfully improve performance”.
>
>   * Our motivation of DROSIA is to demonstrate its effectiveness and efficient in sequence modeling and time series forecasting, so we did not compare it with LLM-based forecasters but with various transformer-based models.
>
> We found some compilation error for the name of a cited paper, so we recompile our paper and have uploaded it again.
>
> If there is any other concern of you, please tell us.
>
> Reference
>
> [1] Are Language Models Actually Useful for Time Series Forecasting? Neurips, 2024

---

### Official Review · Reviewer_eMSN · 2024-11-09

**Soundness:** 2
**Presentation:** 2
**Contribution:** 2
**Rating:** 3
**Confidence:** 4

**Summary:**

In the papers the authors propose a MLP based time series forecasting framework named DROSIA which emphasizes the separate representations of patch (local) level information and sequence level information, i.e. the manual decoupling of the two. DROSIA prefers concatenation instead of summation to explicitly present the related info. They empirically benchmark DROSIA against other SotA methods and demonstrate its outstanding performance.

**Strengths:**

DROSIA as proposed in the paper is a practical architecture to use, and it is relatively convincing that it can deliver good performances.

**Weaknesses:**

There are weaknesses in both the theoretical motivation and the empirical study in the paper. Novelty wise, the motivation and justification behind the decoupling of sequence and patch level information is not well supported, whereas besides this point DROSIA has no outstanding distinctions from other linear (MLP) based model.

For the empirical study there lacks many details regarding, e.g. benchmark model parameters, reasons for setting up the benchmark parameters, etc, which makes the empirical support for the decoupling claim weak.

Presentation wise, the writing could use some revision, specially regarding the key parts, e.g. the algorithm of DROSIA, the reasoning behind the ablation study, etc.

**Questions:**

Regarding the theory:
1. The dot product attention and the skip connection altogether are also capable of decoupling between the patch level presentation and its interaction with the whole sequence even when these two levels of presentations are summed together instead of concatenated. Can you provide a more rigorous or quantifiable definition on what the decoupling means here in the paper?
2. Can you clarify the flow of the algorithm? For example, eq 3 through eq 7, where is S^j_1, assuming C^j is sequence level? How to get S^{j+1}_i?
3. Empirical study reveals no benefit from patching. What's the motivation behind it?

Regarding the empirical study:
1. The choice of DROSIA and other baseline methods' hparams are either unclear or arbitrary across different studies. It would be better to have more details to back a fair comparison, e.g. all methods are tuned to near optimal.
2. The lookback window for each tasks also seem a bit arbitrary, e.g. table 1 uses 96 whereas the original PatchTST paper reports 512. It would be more insightful to report multiple lookup for stronger empirical evidence.
3. Table 4 Effectiveness of DROSIA is too limited in the sense that (1) the benchmark datasets are two and high dimensional, and (2) "S" for PatchTST's original setting is a bit misleading for claiming no patch level presentation there.
4. Based on the current empirical study, it is unclear whether the performance edge in Table 1 is due to the decoupling or due to a better tuning of some linear / MLP structures which are already known to be also effective for forecasting tasks. Consider adding an appendix.

---

> ### Author Response · Authors · 2024-11-14
>
> Thank you for the comments and suggestions. We have carefully considered each point and detail our responses below:
>
> For weaknesses:
>
> We apologize for the impression our paper has given you. Could you please specify which linear (MLP) based models that DROSIA has no outstanding distinctions from, and why you think that "the motivation and justification behind the decoupling of sequence and patch level information is not well supported".
>
> Could you please specify where of the benchmark or model parameters lacks details?
>
> For questions regarding the theory:
>
> 1. This question is great and important for us to clarify our contributions. Please let me explain for this:
>
>   * Firstly, as we mentioned in the introduction, our intuition is from the sociological perspective of Transverse Interaction: Individuals recognize the physical environment as a symbolic other and use this understanding to structure their interaction with a “generalized other” in paper [1]. We understand individuals and collective from this point of view, thus extract the sequential information only once as the "generalized other" for individuals. While the intuition of skip (or residual) connection is to enable training deeper network.
>
>   * The individuals interact with the collective to understand their position in the whole structure, thus enhance their representations. As the repeated process of multi-layer encoder goes, the whole structure will be well built, and the individuals will be in their accurate positions in the structure.
>
>   * As for the dot product attention with the skip connection, it has a quadratic computational complexity. Furthermore, it equips with skip connection not because the understanding that the individual information is compromised in their algorithm. But we emphasize its importance in sequence modeling.
>
>   * We demonstrate the effectiveness and efficiency of DROSIA with extensive experiments and ablation studies, especially in table 4, the forecasting performance of PatchTST (equipped with skip connections) could be further enhanced in "P+S" scenario with even less amount of parameters (embedding dimension from d to d/2), and with the same self-attention to extract the sequential information.
>
>   * As for the reasons, we think, are from that the summed method could not decouple the parameters of the information fusion layer (e.g. FFN), which is very important for balancing the individual and sequential information (they share the same parameter in the summed scenario).
>
>   * Consequently, we believe DROSIA is novel and important to the research fields of time series forecasting and sequence modeling.
>
> 2. As we explained under equation 2: "Equation (2) outlines the overall process of the DROSIA encoder, which will be described in detail from Equation (3) to Equation (7).", our algorithm flows just from equation 3 to equation 7.
>
>   * Above equation 3, we explained: "The output representations from the patch embedding or the previous layer of DROSIA encoder are first concatenated". So S^1 represents the patch embeddings, and S^j denotes the input of the j-th encoder layer, or the output of the (j-1)-th layer. S^j_i is the i-th item of S^j, and the shape of all S is (b * c, n, d/2), b: batches, c: channels, n: patches, d: dimension.
>
>   * We concatenate S^j_i, i=1,2,...,n as C^j (equation 3), the shape of all C is (b * c, n * d/2).
>
>   * We extract sequential information from C^j that is useful for forecasting through any sequence information extractor (equetion 4, we choose MLP just for its simplicity, we have evaluated various methods for extraction in table 5), R^j is the extracted sequential information, the shape of all R is (b * c, d/2).
>
>   * Above equation 5 and equation 6, we explained: "The extracted sequential information is concatenated with the original patch embeddings or the outputs from the previous encoder, as illustrated in Figure 2." We repeat R^j for n times (from (b * c, d/2) to (b * c, n, d/2)) and concatenate it with S^j (equation 5, figure 2), then nomalize the two as D (equation 5, equation 6), the shape of D is (b * c, n, d), finally fuse them to original dimension of S^j to derive S^{j+1}.
>
> 3. The main reason that we patched the input is for a fair comparison between DROSIA and Transformer (employed by PatchTST). And we want to reduce the time consumption of DROSIA.
>
> * We have never claimed that the patch embedding is one of the contributions of this paper.
>
> * Although the patch size has few impact for overall performance of DROSIA, we need experimental results to prove it.
>
> * Since that is the case, we set the patch size to be consistent with PatchTST for fair comparison between DROSIA and Transformer.
>
> For questions regarding the empirical study:
>
> Due to the character limit, we will add another official comment for responces to questions regarding the empirical study.
>
> Reference
>
> [1] Transverse interaction: A pragmatic perspective on environment as other. Symbolic Interaction

---

> > ### Comment · Reviewer_eMSN · 2024-11-25
> >
> > I'd like to thank the authors for the rebuttal. Some details regarding my concerns of the novelty / significance of the paper meeting the ICLR bar in response to "Could you please specify which linear (MLP) based models that DROSIA has no outstanding distinctions from, and why you think that "the motivation and justification behind the decoupling of sequence and patch level information is not well supported".
> >
> > Linear / MLP architectures are ubiquitous and are proven effective for time series forecasting tasks (e.g. NBEATS, NHITS, DLinear, TiDE to name a few), therefore either a revised MLP architecture or a hybrid one ( + transformers) being able to deliver competitive benchmarking performance is, to some extent, not surprising. The only architectural novelty of DROSIA comes from the explicit decoupling. Therefore I think the significance of the paper needs to be supported by either (1) a theoretical justification of any new model capacity introduced (which I don't see but I could've been missing points) or (2) a large empirical improvement over baseline methods.
> >
> > As I mentioned in the review I like the practicality of the paper. Will discuss with other reviewers and AC in the next round.

---

> ### Author Response · Authors · 2024-11-14
>
> For questions regarding the empirical study:
>
> 1. Could you please specify which hparams of the models that our choice "are either unclear or arbitrary across different studies"?
>
> 2. We have reported multiple lookup in ablation study (figure 4), and we set the lookback window for overall experiment to 96 is to stay consistent with the corresponding paper of iTransformer, but we have conducted extensive ablation studies (figure 4) for the impact of various lookback window lengths of DROSIA and other forecasters (including PatchTST). Please let me explain for this ablation study:
>
>     * This ablation study is for the impact of various lookback window lengths {48, 96, 192, 336, 512}. We compared DROSIA with 4 forecasters with different types on various scale datasets.
>
>     * The experimental results (figure 4) show that when longer sequences are input, DROSIA surpasses PatchTST or other forecasters by an even greater margin, which demonstrated the effectiveness of DROSIA for sequence modeling and capturing long-distance dependencies.
>
> 3. We apologize for our introduction to Table 4 makes you confused. Our motivation for DROSIA is to emphasize the importance of individual (patch or point) information, it is like a local-global architecture to enhance sequence modeling. Existing method may sacrifice the individual information when conduct sequence modeling although with the skip (or residual) connection, which has been demonstrated by the experimental results comparison between DROSIA and PatchTST.
>
> 4. We are willing to add an appendix for all of our experimental results. Based on the current empirical study, we believe the effectiveness of DROSIA has been demonstrated, please let me explain:
>
>     * We use the same hyperparameters for DROSIA and all baselines for all experiments and ablation studies, which has been introduced in our paper. And we promise that we will never do "better tuning" only for our proposed models.
>
>     * The only difference between DROSIA and PatchTST is in the encoder, the results in table 1 show that DROSIA surpasses PatchTST for all scenarios, which demonstrates the effectiveness of DROSIA over Transformer. Besides, we will add a statistical analysis for the results in table 1 and ablation studies as soon as possible to prove the effectiveness of DROSIA, and to demonstrate its significance over other forecasters.
>
>     * We propose DROSIA not for an MLP-based forecasters, but an architecture to emphasize the individual information to enhance the sequence modeling, and choose time series forecasting task to demonstrate our motivation.
>
>     * The ablation study (figure 4) shows the better ability for capturing long-distance dependencies of DROSIA over Transformer encoder.
>
>     * The ablation study (table 4) shows that when concatenate the original patch embeddings with self-attention outputs, the forecasting performance of PatchTST would also increase, which demonstrate the effectiveness of our proposed architecture. We only choose ECL and Traffic datasets here not because that the same phenomenon does not exist on small-scale datasets, but that small-scale datasets are more influenced by random factors. Therefore, we have chosen to demonstrate using large-scale datasets only. We are willing to include results on datasets with various sizes in appendix.

---

> ### Author Response · Authors · 2024-11-17
>
> For question 3 regarding the empirical study, we may neglect a point in our previous response for "S for PatchTST's original setting is a bit misleading for claiming no patch level presentation". Please let me explain this.
>
>   * Although PatchTST employs Transformer (with the skip connection) as the encoder, we still concern that the individual information  will be compromised as the two types of information are summed and share the same parameters of the FFN layer, which could not better balance the two.
>
>   * So we regard the original PatchTST as a sequential information only method in the ablation study (table 4), and its performance is further enhanced in the "P+S" scenario.
>
> For question 4 regarding the empirical study, we have conducted 5 trails of DROSIA, iTransformer, PatchTST, and FreTS on 4 ETT subsets with the newest version of TSL repo. All the parameters are used just the same as the scripts we uploaded in supplementary material for all forecasters. The MSE results are reported here, and the same phenomenon is also found for the MAE metric. We checked previous python training file and scripts of TSL, and apologize that we only use 1 encoder layer for all forecasters on small datasets before and report 2 in our paper, and the 5 trials are conducted with the new scripts (2 encoder layers on small datasets) for all models
>
> We will revise our paper to add these results with our analysis, and we are still working for larger datasets and tables in the ablation study. The results could demonstrate the significance of DROSIA over these baselines on small datasets, as the input length increases, DROSIA could surpass all baselines with even greater margin, which should be able to demonstrate the effectiveness of DROSIA in extensive scenarios.
>
> |Model|DROSIA|iTransformer|PatchTST|FreTS|
> |:-:|:-:|:-:|:-:|:-:|
> |ETTh1|0.441 $\pm$ .002|0.454 $\pm$ .001|0.448 $\pm$ .003|0.483 $\pm$ .001|
> |ETTh2|0.379 $\pm$ .002|0.389 $\pm$ .003|0.382 $\pm$ .002|0.531 $\pm$ .025|
> |ETTm1|0.384 $\pm$ .001|0.408 $\pm$ .001|0.388 $\pm$ .002|0.408 $\pm$ .001|
> |ETTm2|0.281 $\pm$ .001|0.291 $\pm$ .001|0.287 $\pm$ .002|0.334 $\pm$ .004|
>
> We will also report the results for larger datasets, other baselines, and ablation studies soon. Is there any other concern of you, please tell us :)

---

> ### Author Response · Authors · 2024-11-26
>
> Thank you for the comments, please let us explain for this.
>
> 1. The novelty of DROSIA is not only in the explicit decoupling, but in introducing the sociological perspective of Transverse Interaction and emphasizing the importance of individual information for sequential modeling. Existing methods, to our best knowledge, do not provide this understanding that the individual information is important to, has been largely compromised in, nor should be in fully interaction with the collective for the sequence modeling process before.
>
> 2. Existing MLP-based forecasters that you have mentioned, such as NBEATS, which is composed of multiple stacks that process the time series in a hierarchical manner, each containing a trend and a seasonality block. NHITS also has a hierarchical structure, which breaks down the time series into multiple levels, each representing a different timescale or resolution. DLinear simply decomposes the original time series into trend and remainder sequences, separately predicts then synthesizes them to obtain the final results. TiDE also decomposes time series data into its underlying components, such as trend, seasonality, and remainder. These models have significant distinctions from DROSIA, which is a non-decomposition model.
>
> 3. We have conducted extensive experiments with many MLP-based models, and DROSIA could significantly surpass them in most of scenarios.

---

### Author Response · Authors · 2024-11-24
**General response to all reviewers**

Dear reviewers,

We have made substantial revisions to our paper, primarily in the following aspects:

1. FreTS is added as a new MLP-based baseline, while Autoformer is removed, due to the too many Transformer-based baselines.

2. We have run many trails to get the average values and standard deviations for all tables in our paper.

3. All of the 8 datasets are evaluated in table 4, and the results are averaged from 4 horizons {96, 192, 336, 720}.

4. The results are also averaged for 4 various horizons in table 5.

We appreciate all the suggestions and comments provided by the reviewers and sincerely thank Reviewer Fnug for the open-mindedness towards our paper, as well as the detailed communication and interaction with us.

As the viewpoint proposed and experimentally demonstrated in our paper, there needs to be sufficient interaction between individuals and the collective to improve the overall structure.

Here, the reviewers represent the ICLR community. As individual authors, we are more than willing to engage in thorough communication with you to enhance our knowledge, thereby better contributing to the collective.

---

> ### Author Response · Authors · 2024-11-29
> **Any other concerns or questions?**
>
> Dear reviewer Mmv4 and XqSP,
>
> Due to the impending end of the rebuttal extension, we would like to know whether our previous responses have addressed your concerns and questions about our paper, or if you have any additional concerns or questions that require our clarification. We are more than willing to engage in thorough communication with you.

---

> ### Author Response · Authors · 2024-12-03
> **Rebuttal period is about to end today**
>
> Dear reviewers,
>
> We appreciate all reviewers for taking time to review our paper and we attach great importance to all comments, questions, or suggestions. We have done our best to address each of these points with detailed responses or substantial revisions, showing a very positive attitude during this rebuttal period.
>
> As the rebuttal period is coming to an end today, we are very eager to know whether our previous responses have addressed your concerns. If there is any other question that require our clarification, please feel free to ask, and we will provide detailed explanations before the author response period ends tomorrow.

---

> ### Author Response · Authors · 2024-12-04
> **Author response**
>
> Dear reviewers,
>
> The rebuttal period has ended. Although we have maintained a positive attitude throughout and have repeatedly requested reviewer Mmv4 and XqSP to inform us whether their concerns have been addressed and if there are any other questions, we have not received any responses to date. We have no other option but to present our last response here, in the hope that the reviewers could take our views into consideration when discussing in the next round.
>
> For Reviewer eMSN,
>
> * It seems that we have addressed most of the concerns, such as the differences between our method and residual connection, the algorithm flow, model structure, experimental setup, and ablation study. Currently, Reviewer eMSN acknowledges the practicality of the paper, but still has concerns regarding the novelty of our work.
>
> * Reviewer eMSN believes that our novelty lies solely in explicit decoupling and considers that our method does not have significant differences from existing MLP-based models, for example, NBEATS, NHITS, DLinear, and TiDE. We have responded to this by providing a detailed explanation of the novelty of our method and the significant differences from the MLP-based models mentioned by the reviewers. Regrettably, we have not received any further response from the reviewer, thus not knowing whether these concerns have been addressed.
>
> For Reviewer Mmv4,
>
> * Reviewer Mmv4 believes that our paper has clarity issues. However, these concerns are not clear nor specific. We have explained the motivation of this paper and the intention behind the settings of model structure again, and asked the reviewer to clearly indicate where these issues exist, but we have not received any response to date, just like all of our responses to the reviewer.
>
> * We have made extensive revisions to the paper, adding citations and discussions for some of the papers mentioned by the reviewer. It seems that the reviewer have a particular interest in transfer learning work. However, since this paper does not involve it, we have not included a discussion on this topic and have explained this many times in our responses.
>
> * Reviewer Mmv4 also mentioned that our model only slightly outperforms PatchTST on small-scale datasets. We have conducted many experiments and added standard deviation comparisons for all experiments. The experimental results show that our model can significantly surpass PatchTST in all scenarios and greatly outperform PatchTST with longer input sequences or on larger-scale datasets.
>
> * However, we are confused by the reviewer's claim that our comparison with channel-dependent models is unfair due to fewer training data. Since we used exactly the same data for all models and have provided a detailed explanation, but we have not received any response from the reviewer.
>
> For Reviewer XqSP,
>
> * Reviewer XqSP also has concerns of the novelty, which has been discussed above. However, the reviewer thought our method has no big difference from PatchTST, which we have discussed in-depth in the methodology section of the paper. Moreover, we have uploaded our codes and scripts in the supplementary material to address the reviewer's concern about the code.
>
> * The lack of in-depth analysis for channel-dependent models is also mentioned, but we have discussed the advantages and disadvantages of channel-dependent models in the related work and experiment section. We have also explained with details why different results of iTransformer in the corresponding paper and our paper.
>
> * Reviewer XqSP suggests that we should conduct more analysis like Figure 10 in iTransformer rather than only computational complexity. However, our method could surpass all channel-independent baselines in most of scenarios, surpass all baselines on most of small datasets, and surpass channel-dependent baselines on large datasets with a fair input length (not too short when compared to the channels) with only linear complexity. We believe these results could demonstrate the effectiveness and efficiency of our method.
>
> * However, We have not received any response from Reviewer XqSP to date.
>
> For Reviewer Fnug,
>
> * We have addressed the concerns of Reviewer Fnug through detailed communication and interaction, in which we have learned a lot not only as authors but also as reviewers for ICLR. We sincerely thank the reviewer for the open-mindedness and positive attitude towards we individual authors.

---

### Note · Authors · 2025-01-23

I have read and agree with the venue's withdrawal policy on behalf of myself and my co-authors.